# Nanotransistor-based gas sensing with record-high sensitivity enabled by electron trapping effect in nanoparticles

Qitao Hu[1,4], Paul Solomon[2], Lars Österlund [3] & Zhen Zhang [1] ✉

Highly sensitive, low-power, and chip-scale $H_2$ gas sensors are of great interest to both academia and industry. Field-effect transistors (FETs) functionalized with Pd nanoparticles (PdNPs) have recently emerged as promising candidates for such $H_2$ sensors. However, their sensitivity is limited by weak capacitive coupling between PdNPs and the FET channel. Herein we report a nanoscale FET gas sensor, where electrons can tunnel between the channel and PdNPs and thus equilibrate them. Gas reaction with PdNPs perturbs the equilibrium, and therefore triggers electron transfer between the channel and PdNPs via trapping or de-trapping with the PdNPs to form a new balance. This direct communication between the gas reaction and the channel enables the most efficient signal transduction. Record-high responses to 1–1000 ppm $H_2$ at room temperature with detection limit in the low ppb regime and ultra-low power consumption of ∼300 nW are demonstrated. The same mechanism could potentially be used for ultrasensitive detection of other gases. Our results present a supersensitive FET gas sensor based on electron trapping effect in nanoparticles.

Hydrogen gas ($H_2$) is one of the most promising candidates for clean and renewable energy sources, replacing fossil fuels towards a future hydrogen society[1–5]. However, any small leakage of $H_2$ during its production, transportation, storage, and usage can be dangerous, due to its ultra-small molecule size, colorless and odorless nature, low ignition temperature, and wide explosive concentration range[6]. Hydrogen safety demands highly sensitive $H_2$ sensors with short response time to detect hydrogen leakage at concentrations far below its explosion limit[7–10]. Additionally, low-power consumption[11–13] and miniaturized device size[14,15] are desirable for remote and distributed $H_2$ sensors.

Palladium nanoparticles (PdNPs) decorated field-effect transistors (FETs) have emerged as a promising device candidate for $H_2$ sensing[16–22]. The channel current of an FET-based $H_2$ sensor is modulated by the work function change induced by $H_2$ reaction with PdNPs, which proceeds via $H_2$ dissociation and subsequent absorption of H atoms in Pd[23]. These devices have gained considerable attention owing

to the current amplification in the FET, integration compatibility with CMOS circuitry, device miniaturization, and mass production possibility[24]. Besides, the large surface-to-volume ratio of dispersed PdNPs can enhance the sensitivity and enable room-temperature sensing capability[25,26]. However, in these devices, any solid gate covering the PdNPs will inevitably block their accessibility to $H_2$. Therefore, an alternative gate, normally a back (substrate) gate is needed to turn on a buried channel and set the working point of the FET. Consequently, the PdNPs are electrically floating, thus decoupled from the gate-channel loop. This leads to indirect capacitive coupling between the PdNPs and the FET channel, thus yielding weak signal transduction. Different novel FET device designs, e.g., ultra-thin planar channel[15,27] and nanowire channel with intimately attached side-gates configuration[28], were recently demonstrated to enhance such capacitive coupling by reducing the distance of the sensing layers from the channel. However, the intrinsic problem of indirect capacitive

[1]Division of Solid-State Electronics, Department of Electrical Engineering, Uppsala University, BOX 65, SE-75121 Uppsala, Sweden. [2]IBM T. J. Watson Research Center, Yorktown Heights, NY 10598, USA. [3]Division of Solid-State Physics, Department of Materials Science and Engineering, Uppsala University, BOX 35, SE-75103 Uppsala, Sweden. [4]Present address: Department of Radiology, Stanford University, Stanford, CA 94305, USA. ✉e-mail: zhen.zhang@angstrom.uu.se

coupling induced by the floating PdNPs sensing layer has not been addressed to date.

In this work, we address the fundamental issue of indirect capacitive coupling of PdNP-FET $H_2$ sensors by using a nanoscale FET sensor design to enable a signal transduction mechanism based on electron trapping effect in PdNPs. In this design, an oxide passivated silicon nanowire (SiNW) channel is gated by two side-gates via nanoscale air gaps (NAGs)—SiNW-NAG FET. The NAGs allow PdNPs to be deposited at the closest possible location to the conducting channel forming on the SiNW sidewalls. An ultra-thin passivation oxide (2 nm) enables electrons to tunnel between the channel and the PdNPs and equilibrate them. The PdNPs work as both $H_2$ sensing material and electron traps. When exposed to $H_2$ gas, $H_2$ can access and react with the PdNPs with fast kinetics. The PdNP-$H_2$ reaction will change the potential energy of the PdNPs thus perturbing the equilibrium between the channel and the PdNPs. As a result, electrons will transfer between the channel and the PdNPs via trapping or de-trapping with the PdNPs to reach a new balance, leading to a steady change in the channel current. The direct electron tunneling communication between the channel and PdNPs enables the most efficient signal transduction. Based on this mechanism, record-high responses to 1–1000 ppm $H_2$ are demonstrated, showing $4.86 \times 10^6\%$ channel resistance change to 1000 ppm $H_2$ at room temperature. A sensitivity of 3600%/ppm and lower limit of detection (LOD) of 4.4 ppb can be deduced. Due to the small size of the SiNW channel, a power consumption of $\sim 300$ nW is sufficient to drive the sensor. Selectivity against three different types of gases, CO (neutral), $NO_2$ (electrophilic), and $NH_3$ (nucleophilic) are also verified. The demonstrated signal transduction mechanism could potentially be used for detecting other gases using different sensing NPs.

## Results

### SiNW-NAG FET structure

A three-dimensional (3D) schematic of the SiNW-NAG FET sensor design is shown in Fig. 1a. The device was fabricated on a (100) silicon-on-insulator (SOI) wafer with 145 nm thick buried $SiO_2$ (BOX) layer using standard Si process technology. The source (S) and drain (D) regions of the FET were heavily $p$-doped (brown), while the SiNW channel consists of a 600 nm long lightly doped $p$-type region (gray). $SiO_2$ passivation layers, with different thicknesses, were deposited surrounding the SiNW via the atomic layer deposition (ALD) process. Side-gates of NiSi were formed in proximity to the SiNW sidewalls, leaving 50 nm wide air gaps in between as dielectric layers. PdNPs were deposited on the top and sidewalls of the $SiO_2$-passivated SiNW by depositing a thin layer (<1 nm) of Pd using electron beam evaporation, which yields agglomerated PdNPs similar to previous work[29]. To deposit the PdNPs on the SiNW sidewalls, double angled depositions were made with the tilted substrate. A detailed description of the fabrication process can be found in the Methods Section and step-by-step process schematics in Supplementary Section 1. Back-gate SiNW FETs, without any side-gates and with PdNPs only on the top of the SiNW, were fabricated in parallel.

Scanning electron microscope (SEM) images of the SiNW-NAG FET device with 4 nm thick passivation $SiO_2$ are shown in Fig. 1b (top) and Supplementary Section 2, respectively. The cross-sectional transmission electron microscope (TEM) image and its element analysis of the PdNP-decorated SiNW (see Fig. 1b, bottom) confirm the SiNW dimension (height: 25 nm, width: 35 nm), the uniform $SiO_2$ passivation layer and the deposition of PdNPs. The voltage applied to the side-gates ($V_{G\_s}$) can effectively modulate the channel conductance via NAG capacitance ($C_{NAG}$), as evidenced by the simulated potential distribution in Supplementary Section 3 and the measured typical transfer and output characteristics in Supplementary Section 4.

### Side and back-gate capacitive coupling

Hydrogen dissociates and populates interstitial lattice sites when it adsorbs on Pd and diffuses into the bulk to form Pd hydride. H (donor) hybridizes with Pd (acceptor) and modifies the electronic properties of Pd (the H1$s$-Pd4$d$ bonding), leading to a detectable shift of its work function. The formation and dissociation of palladium hydride are reversible processes. The work function of Pd is sensitive to changes in hydrogen across a wide range of concentrations, which allows for precise measurement of hydrogen concentration. To investigate the Pd-based $H_2$ transistor sensors, semiconductor device simulator provides a powerful tool[30,31]. For simulation and modeling purposes, the sensing mechanism of a Pd sensor is typically described by the formation of a dipole layer as the interface becomes polarized[18,28,32] (see inset Fig. 1c). The first advantage of the SiNW-NAG FET structure in signal transduction is illustrated in the equivalent circuits in Fig. 1c. In a conventional back-gate FET (see Fig. 1c left), the work function change in PdNPs ($\Delta\Phi_{PdNP}$) caused by their reaction with $H_2$ is weakly coupled to the top channel in the SiNW via small stray capacitance ($C_{stray}$; $C_{Si}$ is the capacitance of Si channel and $C_{stray} \ll C_{Si}$), as shown in the red loop in Fig. 1c left. Only a small portion of $\Delta\Phi_{PdNP}$ is coupled to the top channel, leading to an insignificant variation of carrier density in the top channel. In addition, the main channel is formed at the bottom side of the SiNW by the back-gate via a strong coupling loop over the BOX layer (see the black loop; $C_{BOX}$ is the capacitance of the BOX layer, and $V_S$ and $V_{G\_b}$ are the voltages applied on the source and back-gate, respectively). Since the main channel, having many more conducting carriers than the top channel is not significantly affected by $\Delta\Phi_{PdNP}$, the overall current change induced by PdNP-$H_2$ reaction will therefore be diluted by the main channel. As a direct contrast, in the SiNW-NAG FET, $\Delta\Phi_{PdNP}$ is modified within the SiNW-NAG-gate coupling loop which generates the main conducting channel on the SiNW sidewalls (Fig. 1c right). $\Delta\Phi_{PdNP}$ is therefore directly coupled to the main conducting channel via $C_{NAG}$. Since $C_{NAG}$ is much larger than $C_{stray}$, a much larger portion of $\Delta\Phi_{PdNP}$ is coupled to the main channel thus producing significantly enhanced modulation of its conductance.

Previous studies have demonstrated that the coupling between the sensing material and the FET channel could be enhanced by using an ultra-thin channel[15,27]. However, the indirect coupling remains in the back-gate FET device structure. SiNW channel with intimately attached side-gates in a FET gas sensor has been reported as an alternative device design to enhance the signal transduction. Nevertheless, in this device design, the sensing PdNPs are out of the channel-gate loop[28]. Consequently, direct $\Delta\Phi_{PdNP}$ coupling to the channel could not be achieved either. Such indirect coupling results in a limited signal transduction efficiency and thereby a minor sensitivity.

The $H_2$ sensing measurement results further prove the enhanced sensitivity of the SiNW-NAG FET device over the control device with the traditional back-gate structure. Both devices were passivated with 4 nm thick $SiO_2$ layers. The measurements were done at room temperature with no external heating. The drain voltage was fixed at $V_D = 1$ V and the drain-to-source current ($I_{DS}$) baseline was stabilized at $\sim 300$ nA with fixed gate voltage setting, leading to a standby power consumption of $\sim 300$ nW. Figure 1d, e shows the comparison of the real-time monitoring of $I_{DS}$ of the traditional back-gate FET sensor and the SiNW-NAG FET sensor, respectively. Both devices exhibit reversible current responses when exposed to the pulses of varied $H_2$ concentration. The response amplitude is defined as the percent change in channel resistance $\Delta R/R_0$, which is shown in Supplementary Section 5. The $H_2$ sensitivity is 1.71%/ppm ($\Delta R/R_0$ per ppm) for the SiNW-NAG FET sensor, which is significantly larger than the back-gate FET (0.035%/ppm). The simulation results also show a higher response of the SiNW-NAG FET device (see Supplementary Section 6). In addition, the LOD of the SiNW-NAG FET sensor with 4 nm passivation $SiO_2$ is extrapolated to be 2.7 ppm, much lower than the back-gate counterpart (48 ppm) (see Supplementary Section 5 for more details).

## Electron trapping effect in PdNPs

When the thickness of the passivation $SiO_2$ layer is reduced to 2 nm, the subthreshold slope (*SS*) of the SiNW-NAG FET exhibits significant degradation, as shown in Fig. 2a. This indicates an increased density of traps in proximity to the SiNW surface with the thin $SiO_2$ layer, which is accessible to electrons in the channel[33]. To nail down the origin of the traps, we fabricated and characterized a PdNP-free SiNW-NAG FET device with 2 nm $SiO_2$ layer (see Fig. 2b). It exhibits a significantly sharper *SS* compared to the PdNP-decorated counterpart. This indicates that the PdNPs act as the electron traps in the 2 nm $SiO_2$ layer passivated SiNW-NAG FET. Electrons in the channel can tunnel through

the 2 nm $SiO_2$ layer and get trapped in the PdNPs. Such tunneling process is suppressed with the 4 nm $SiO_2$ layer, which explains the *SS* dependence shown in Fig. 2a. On the other hand, the dynamic electron trapping/de-trapping processes with the PdNPs are expected to generate extra low-frequency noise (LFN)[34]. Indeed, the device with the thinner oxide exhibits higher LFN (see Supplementary Section 7), which double confirms the electron trapping effect in the PdNPs.

Due to the small size of the PdNPs (2–3 nm in diameter, as measured by TEM, see Fig. 1b), a strong Coulomb blockade exists. Each PdNP can only trap one electron and generate a single energy state ($E_{PdNP}$) since the entry of the 2nd electron is not energetically

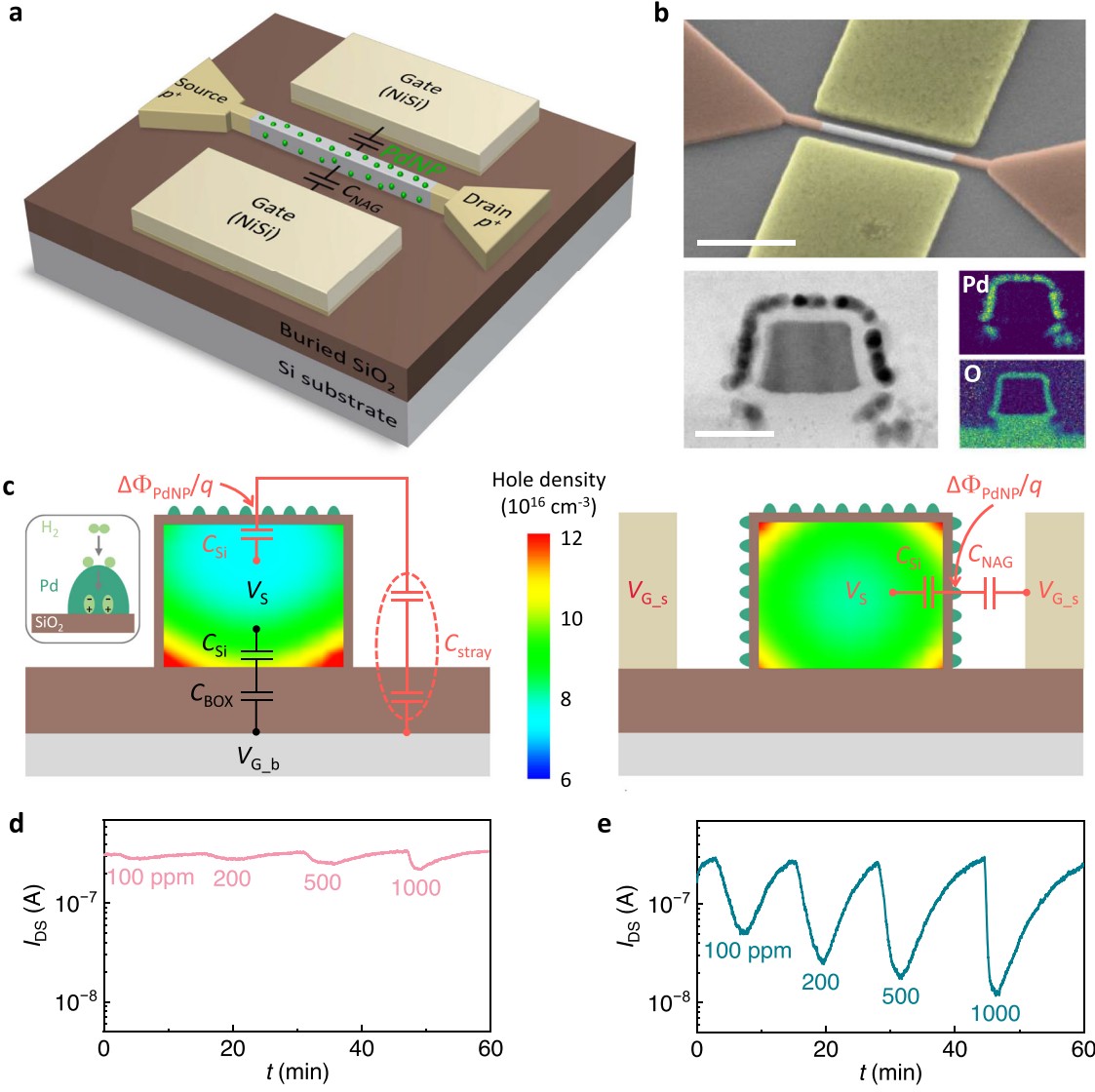

**Fig. 1 | Device structure of the silicon nanowire gated via nanoscale air gaps field-effect transistor (SiNW-NAG FET) and $H_2$ sensing tests based on conventional capacitive coupling mechanism. a** Three-dimensional schematic of the SiNW-NAG FET device for $H_2$ detection. The SiNW channel is *p*-type and gated by the side-gates via nanoscale air gaps (NAGs). The $H_2$ sensing layer palladium nanoparticles (PdNPs) were deposited within a coupling loop between the side-gates and the main channel. **b** Three-dimensional scanning electron microscope (SEM) image of the SiNW-NAG FET (top; scale bar, 400 nm) and cross-sectional transmission electron microscope (TEM) image (bottom-left; scale bar, 20 nm) and color-coded energy dispersive X-ray spectroscopy (EDS) images (bottom-right) of the PdNP-decorated SiNW channel. The SEM image is colorized; the brown, green, lightly grey, and heavily grey areas represent the heavily doped source/drain, NiSi side-gates, lightly doped channel, and substrate, respectively. The PdNPs cover

both the top surface and sidewalls of the SiNW with a 4 nm thick $SiO_2$ passivation layer in between. The green area in the EDS images indicates the corresponding element distribution. **c** Analysis of the capacitive coupling of the $H_2$ signal in the conventional back-gate SiNW FET (left) and the SiNW-NAG FET (right). The simulated cross-sectional distribution of holes was obtained at drain voltage $V_D = 1$ V and fixed working point of drain-to-source current $I_{DS} = 300$ nA by proper gate voltage setting. The red loops in both devices indicate the gas reaction coupled capacitor loops, which generate the current responses in the SiNW. The black one in the back-gate SiNW FET (left) refers to the $H_2$-insensitive loops. Real-time $H_2$ sensing results measured at room temperature of (**d**) the back-gate SiNW FET and (**e**) the SiNW-NAG FET. Both devices are passivated with 4 nm $SiO_2$. $I_{DS}$ sampling was performed in a series of $H_2$ pulses at concentrations ranging from 100 to 1000 ppm.

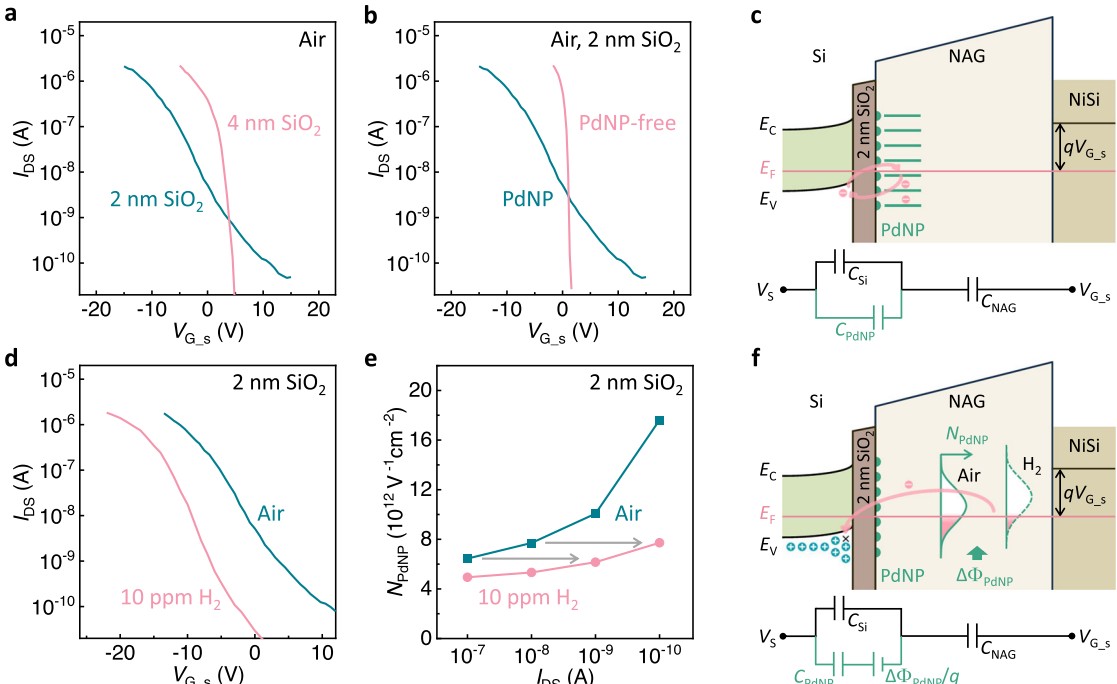

**Fig. 2 | Electron trapping effect in palladium nanoparticles (PdNPs). a** Transfer curves of the silicon nanowire gated via nanoscale air gaps field-effect transistors (SiNW-NAG FETs) with 2 and 4 nm thick SiO$_2$ passivation layers. **b** Transfer curves of the 2 nm SiO$_2$ layer passivated SiNW-NAG FET with and without PdNP decoration. All curves in (**a**) and (**b**) were measured in air. **c** Schematic and equivalent circuit of the SiNW-NAG FET with 2 nm SiO$_2$ passivation layer in air. The PdNPs work as electron traps. The trapping/de-trapping processes are enabled via the electron tunneling through the thin SiO$_2$ layer. $E_C$, $E_V$, $E_F$, and $V_S$ are the bottom of conduction band energy, top of valence band energy, Fermi level, and source voltage, respectively. **d** Transfer curves of the 2 nm SiO$_2$ layer passivated SiNW-NAG FET in air and 10 ppm H$_2$. **e** Extracted density-of-state of the PdNPs from (**d**) as a function of channel current in air and 10 ppm H$_2$. **f** Schematic and the equivalent circuit of the SiNW-NAG FET with 2 nm SiO$_2$ passivation layer in H$_2$ ambience. Once the PdNP energy is raised above $E_F$ due to the reaction with H$_2$, the electrons trapped in the corresponding PdNPs are released back to the SiNW channel and recombine the holes in it. $N_{PdNP}$ and $\Delta\Phi_{PdNP}$ are the density-of-states and work function change of PdNPs.

favorable[35]. Considering the size distribution of the PdNPs and the monotonic size-dependence of nanoparticle's work function[36], the energy states of the PdNPs are distributed in energy space, as illustrated in Fig. 2c. With the 2 nm thick SiO$_2$ layer, electrons can tunnel between the SiNW channel and the energy states of PdNPs and equilibrate them. Therefore, each PdNP with energy lower than Fermi level ($E_F$) in the channel will trap an electron from the channel, leaving the remaining PdNPs empty. The trapping effect in the PdNPs contributes an equivalent capacitor ($C_{PdNP}$), which is in parallel to the Si channel capacitor $C_{Si}$ (see Supplementary Section 8 for the detailed process to develop the equivalent circuit). $C_{PdNP}$ on 2 nm SiO$_2$ layer is estimated to be 20 times of $C_{Si}$ based on the significant $SS$ change induced by the presence of the PdNPs (see Supplementary Section 9 for detailed analysis).

As shown in Fig. 2d, exposure to H$_2$ shifts the transfer curves of the SiNW-NAG FET with 2 nm SiO$_2$ towards lower gate voltage and sharpens $SS$. These effects could be explained by a simple H$_2$-mediated energy shift of the PdNP trap distribution. $SS$ depends on the density-of-states of the PdNPs ($N_{PdNP}$) close to $E_F$; states below $E_F$ are already filled with electrons, while electrons do not have enough energy to access the PdNP states far above $E_F$. Thus, $N_{PdNP}$ can be extracted from $SS$ (see Fig. 2e)[33], which increases at lower $I_{DS}$ (higher $E_F$ in the $p$-type channel) in both air and H$_2$. Such non-uniform distribution of $N_{PdNP}$ can be explained by the size-dependence of PdNP's work function. It is known that the evaporator-deposited metal nanoparticles exhibit a Gaussian size distribution[37]. Since nanoparticle's work function exhibits a monotonic size dependence, a Gaussian-shape energy distribution of the PdNPs can be expected (see Fig. 2f). The increased $N_{PdNP}$ at lower $I_{DS}$ indicates that $E_F$ falls in the lower half region of the PdNP density profile. Moreover, Fig. 2e shows a decreased $N_{PdNP}$ when

exposed to H$_2$, which can be attributed to the globally raised PdNP density profile in energy space, owing to the PdNP-H$_2$ reaction induced work function change. The reduced $N_{PdNP}$ in H$_2$ is double confirmed with the lower noise measured in H$_2$ ambience (see Supplementary Section 7) since the noise generated by the dynamic electron trapping/de-trapping processes with the PdNPs is also determined by $N_{PdNP}$ that the channel electrons can access. It is noticeable in Fig. 2e that $N_{PdNP}$ at $I_{DS} = 10^{-9}$ and $10^{-10}$ A in 10 ppm H$_2$ can be obtained from that in air via roughly two orders of magnitude $I_{DS}$ shifting (which is equivalent to $\sim$120 mV change in $E_F$, since the carrier density in the channel follows Boltzmann relationship with $E_F$), suggesting that the PdNP density profile is raised by $\sim$120 mV upon exposure to 10 ppm H$_2$.

**Super-sensitivity enabled by the electron trapping effect**

Hydrogen absorption in PdNPs will raise $E_{PdNP}$, thus perturbing the equilibrium between the channel and PdNPs. This will force PdNPs to detrap electrons to the channel until a new balance is reached. Specifically, once $E_{PdNP}$ exceeds $E_F$, the electrons trapped in the corresponding PdNPs are expected to spill and flow back to the SiNW channel and recombine with the holes to generate a current response, as illustrated in Fig. 2f. Owing to the large capacitance of $C_{PdNP}$, the number of the released electrons and the resultant current response could be significant. The electron trapping/de-trapping processes in the SiNW-NAG FET sensor generate direct communication between the PdNP-H$_2$ reaction and the main conducting channel, thus enabling the most efficient signal transduction. The weak capacitive coupling and the signal dilution issues in the back-gate FET gas sensors are therefore avoided.

The H$_2$ sensitivity based on the electron trapping mechanism was tested using the SiNW-NAG FET with 2 nm SiO$_2$ passivation layer at

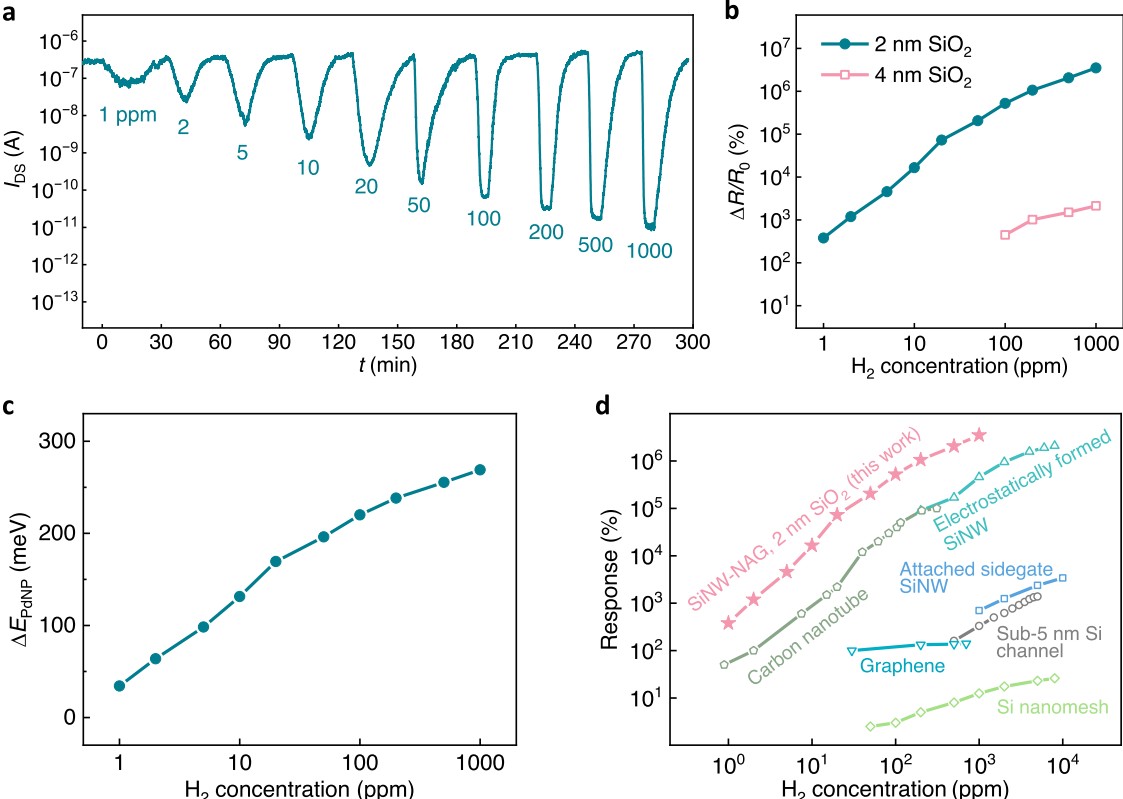

**Fig. 3 | $H_2$ responses of 2 nm $SiO_2$ layer passivated silicon nanowire gated via nanoscale air gaps field-effect transistor (SiNW-NAG FET). a** Real-time $H_2$ sensing results at varied concentration at room temperature. **b** Percent change in resistance $\Delta R/R_0$ of the SiNW-NAG FET devices with 2 and 4 nm thick $SiO_2$ passivation layer as a function of $H_2$ concentration. $\Delta R/R_0$ with 2 and 4 nm $SiO_2$ layers is extracted from (**a**) and Fig. 1e, respectively. **c** Calculated energy variation of palladium nanoparticle $\Delta E_{PdNP}$ as a function of $H_2$ concentration from (**b**). **d** Benchmark of $\Delta R/R_0$ of the 2 nm $SiO_2$ layer passivated SiNW-NAG FET with the corresponding reported data of FET $H_2$ sensors: Si nanomesh[17], electrostatically formed SiNW[18], carbon nanotube[21], attached sidegate SiNW[28], sub-5 nm Si channel[27], and graphene[52]. All devices were functionalized with Pd or its alloys and the measurements were performed at room temperature.

room temperature. Figure 3a shows the real-time $I_{DS}$ monitoring upon exposure to varied $H_2$ concentrations ranging from 1 to 1000 ppm. The resistance change $\Delta R/R_0$ is extracted and plotted in Fig. 3b. The device shows ultrahigh room-temperature responses, which are exemplified with $\Delta R/R_0 = 4.86 \times 10^6\%$ at 1000 ppm $H_2$ and $\Delta R/R_0 = 4.02 \times 10^2\%$ at 1 ppm $H_2$. The reproducibility of the device is demonstrated in Supplementary Section 10. Also shown in Fig. 3b, $\Delta R/R_0$ with 2 nm $SiO_2$ layer is roughly three orders of magnitude higher than that with 4 nm $SiO_2$ layer in the 100–1000 ppm $H_2$ range. A greatly enhanced sensitivity (3600%/ppm with 2 nm $SiO_2$, see Supplementary Section 5) is also demonstrated compared with the 4 nm $SiO_2$ layer passivated device (1.71%/ppm). This confirms the overwhelming sensitivity enabled by the electron trapping effect in the PdNPs. In the device with 4 nm $SiO_2$, the thick $SiO_2$ layer blocks the electron tunneling process, so only a small portion of the work function change is coupled to the channel via capacitive coupling ($\Delta E_F \ll \Delta E_{PdNP}$). In contrast, for the 2 nm $SiO_2$ passivated device, electron trapping/de-trapping processes via tunneling through the thin $SiO_2$ layer enable direct communication between the channel and the PdNPs. Consequently, $E_F$ in the SiNW closely adapts to $E_{PdNP}$, leading to $\Delta E_F$ close to $\Delta E_{PdNP}$, which is much higher than $\Delta E_F$ achieved via capacitive coupling. The quantitative analysis of $\Delta E_F$ with the thin $SiO_2$ passivation layer is available in Supplementary Section 11. Higher $\Delta E_F$ induced by the electron trapping effect results in a significantly enhanced current response. It is notable that the LOD of 2 nm $SiO_2$ passivated SiNW-NAG FET sensor is extrapolated to be 4.4 ppb (see Supplementary Section 5), which is more than two orders of magnitude lower than that with 4 nm $SiO_2$ (2.7 ppm), and significantly lower than the previously reported FET[18] and optical[9] $H_2$ sensors. It should be emphasized that this signal

transduction mechanism demonstrated for $H_2$ sensing in this work could also be used for other target gases, provided the work function of the sensing NPs can be changed upon exposure to the target gas. For instance, Au NPs for ozone[38] and $SnO_2$ NPs for $NO_2$[39].

Since $\Delta E_F$ in the channel approximates to $\Delta E_{PdNP}$, it is reasonable to estimate the latter using $\Delta E_F$, which can be extracted from the responses in Fig. 3b using the Boltzmann relationship. The $H_2$-induced $\Delta E_{PdNP}$ extracted via the electron trapping mechanism (see Fig. 3c) is comparable with the reported data[40]. Specifically, $\Delta E_{PdNP}$ at 10 ppm $H_2$ is estimated to be 131 mV, which is consistent with the analysis of Fig. 2e. The ultrahigh responses demonstrated with the signal transduction mechanism are benchmarked with recently reported FET-based $H_2$ sensors that were functionalized with Pd or its alloys and measured at room temperature, which is compiled in Fig. 3d. The responses demonstrated in our work are the highest among these devices in the $H_2$ concentration range from 1 ppm to 1000 ppm.

## Kinetics and selectivity

Apart from the sensitivity discussed above, fast response and recovery and good selectivity are also critical for a high-performance gas sensor. As shown in Fig. 4a, the response time of the SiNW-NAG FET sensors with 4 and 2 nm thick $SiO_2$ layer, defined as the time for realizing 90% current response ($t_{90}$), is extracted from the real-time current traces in Figs. 1e and 3a, respectively. $t_{90}$ is almost the same for both sensors based on the electron trapping mechanism (2 nm $SiO_2$) and the conventional capacitive coupling mechanism (4 nm $SiO_2$). This confirms that the response to $H_2$ is not slowed down with the electron trapping mechanism, which indicates that the electron tunneling process through the 2 nm $SiO_2$ layer is much faster than the PdNP-$H_2$ reaction.

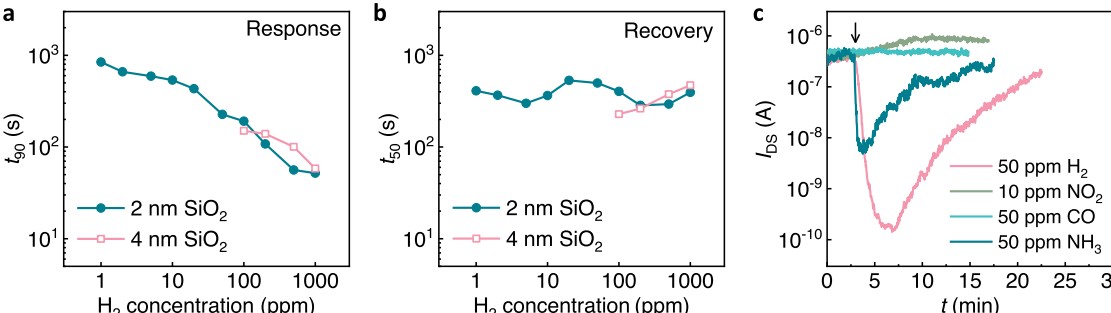

**Fig. 4 | H$_2$ sensing kinetics and selectivity. a** Response time $t_{90}$ and (**b**) recovery time $t_{50}$ of the silicon nanowire gated via nanoscale air gaps field-effect transistor (SiNW-NAG FET) devices with 2 and 4 nm SiO$_2$ layers. $t_{90}$ and $t_{50}$ with 2 and 4 nm SiO$_2$ layer are extracted from Figs. 3a and 1e, respectively. **c** Current responses to 50 ppm H$_2$, 50 ppm CO, 10 ppm NO$_2$, and 50 ppm NH$_3$ in synthetic air of the SiNW-NAG FET device with 2 nm SiO$_2$ layer.

The longer response time at lower H$_2$ concentration can be attributed to the low sticking probability of H$_2$ on Pd which limits the H$_2$ reaction[41]. In addition, the recovery time, which is the time for recovering 50% of current response ($t_{50}$), is not prolonged with the tunneling process either (see Fig. 4b). The response and recovery processes could be further accelerated by integrating a micro-heater in proximity to the H$_2$ sensor[15] or suspending the SiNW for enhanced Joule heating in the SiNW[42,43].

The SiNW-NAG FET device can readily discriminate H$_2$ from interfering electrophilic, neutral molecules, and nucleophilic molecules. Here, CO (neutral), NO$_2$ (electrophilic), and NH$_3$ (nucleophilic) gas molecules were selected as interfering gases to demonstrate the selectivity. The responses with 2 nm SiO$_2$ layer to 50 ppm CO, 10 ppm NO$_2$, and 50 ppm NH$_3$ diluted in synthetic air are presented in Fig. 4c, which are significantly lower than the response to 50 ppm H$_2$. The insensitivity towards CO is due to the absence of charge transfer of the neutral CO molecule. The sensor response towards NO$_2$ even results in an opposite current response and is likely due to NO$_2$ adsorption on the channel walls. NH$_3$ exposure under our conditions (synthetic gas at low catalyst temperature) is known to lead to NH$_3$ dissociation and PdN$_x$ formation with small amounts of N$_2$ and H$_2$O products, and very few NOx species[44]. A dipole layer can then be created by NH$_3$ exposure (similar to H$_2$), yielding a work function change, albeit much smaller than for H$_2$, and hence an almost 100 times smaller signal (Fig. 4c). Finally, we note that the SiNW-NAG FET device structure with the electron trapping based signal transduction mechanism is compatible with most of the mitigation approaches developed to enhance sensors' immunity to cross-interferences, such as alloying the NPs[45] or using a filtering film[46,47].

The high performance of our SiNW-NAG FET H$_2$ sensor is promising in various applications. For instance, it could be used for H$_2$ safety monitoring in buildings and process industry owing to the low energy consumption, high sensitivity, low cost, and easy integration[48]. Besides these characteristics, the selectivity against CO makes our sensor a good candidate for H$_2$ leakage detection in H$_2$-powered vehicles, considering the possible interference with CO released from other combustion engines[49]. To realize real-world applications, long-term stability needs to be ensured, e.g., by covering the PdNPs with a polymer layer[46], as well as admissible working temperatures and humidity conditions.

## Discussion

In summary, we demonstrate supersensitive nanotransistor-based gas sensing enabled by the electron trapping effect in nanoparticles. The sensor device is a SiNW FET fabricated using a CMOS-compatible process. The SiNW channel passivated with a thin SiO$_2$ layer is gated by two side-gates via NAGs. The NAGs allow the sensing NPs to be deposited on the sidewalls of the SiNW so that the conducting channels are generated at the closest possible location to the NPs. Target gas can access the NPs via the air gaps with no physical barrier. When the SiO$_2$ layer is thin enough, electrons in the main channel can tunnel between the main channel and the sensing NPs and equilibrate them. The gas reaction with the NPs will perturb the equilibrium, thus forcing electron transfer between the channel and the NPs via electron trapping/de-trapping processes, which generate the current signal. The electron trapping effect in the NPs provides direct communication between gas reaction and the main conducting channel thus enabling the most efficient signal transduction. We demonstrate a record-high H$_2$ sensitivity of 3600%/ppm and ultra-low LOD of 4.4 ppb in this type of side-gate FET H$_2$ gas sensor device at room temperature and with ultra-low power consumption of around 300 nW. The SiNW-NAG FET device could potentially be used for detecting other gases.

## Methods
### Device fabrication
The SiNW FET sensors were fabricated in 100-mm SOITEC SOI wafers with standard Si process technology[50]. The SOI wafers are composed of a 55 nm thick lightly doped $p$-type Si layer on the top of a 145 nm thick buried oxide layer. The top Si layer was thinned down from 55 to 30 nm by thermal oxidation and subsequent oxide etching in HF acid. BF$_2$ implantation was used to form the heavily $p$-doped ($p^+$) side-gate (G), source (S), and drain (D) regions, while the channel region was protected by electron-beam resist UVN during the implantation. The dopants in S/D were activated by rapid thermal processing at 1000 °C for 10 s in the N$_2$ atmosphere. The SiNW channel with S/D contacts and the side-gates were defined by electron-beam lithography (EBL) with 2% hydrogen silsesquioxane (HSQ) resist, followed by reactive ion etching (RIE). Subsequently, a layer of SiO$_2$ was deposited on the device via ALD with a pre-cleaning step in piranha solution for 15 min for chemical oxide growth. A 20 nm thick nickel silicide layer was formed on both G and S/D regions via 30 s rapid thermal annealing at 400 °C. PdNPs were deposited on the SiO$_2$-passivated SiNW through electron beam evaporation process with 0.1 Å/s rate and 5 Å thickness. The substrate was tilted at 60° and double deposition was used to enable the PdNP deposition on both sidewalls of the SiNW. A back-gate SiNW FET with device, the side gates were not defined at EBL step and only the top surface of the SiNW was functionalized with the PdNPs (deposition without substrate tilt). The thickness of the Pd layer was kept constant for both devices.

### Electrical measurements
All electrical measurements were performed in Linkam LTS420E-P chamber at room temperature. Transfer ($I_{DS}$ versus $V_G$) and output ($I_{DS}$ versus $V_D$) characteristics and real-time current sampling ($I_{DS}$ versus $t$) were measured using an HP4155A semiconductor parameter analyzer. The PSD of $I_{DS}$ was characterized using a Keysight E4727A advanced low-frequency noise analyzer. For each measurement, $V_{DS}$ was biased at 1 V. The varied gas concentration in the gas sensor measurements

was done by diluting the analyte in synthetic air ($O_2:N_2 = 1:4$) and the total gas flow was fixed at 500 sccm. The sources of $H_2$, CO, $NO_2$, and $NH_3$ were all research-grade gases from Air Liquide diluted in $N_2$ with the following concentrations: 5% $H_2$, 100 ppm CO, 100 ppm $NO_2$, and 100 ppm $NH_3$, respectively. The gases were subsequently diluted to desired target analyte concentrations by mixing with synthetic air by means of mass flow regulators.

## Device simulation
The commercially available Sentaurus TCAD device simulator (version: S-2021.06-SP1) is used for the FET simulation[51]. Mobility models include doping dependence, high-field saturation, and transverse field dependence. The intrinsic carrier concentration is determined with the silicon bandgap narrowing model OldSlotBoom. Shockley–Read–Hall model with a doping-dependent lifetime is used for simulating recombination process.

## Reporting summary
Further information on research design is available in the Nature Portfolio Reporting Summary linked to this article.

## Data availability
The data that support the findings of this study are available within the paper and its supplementary information files. Source data are provided with this paper.

## Code availability
The custom codes used for the device simulations are available from the corresponding author upon request.

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

## Acknowledgements

This work was supported by Swedish Strategic Research Foundation FFL15-0174 (Z.Z.), Swedish Research Council VR2014-05588 and VR2019-04690 (Z.Z.), Wallenberg Academy Fellow KAW2015-0127 and its extension KAW2020-0190 programs (Z.Z.), H2020-MSCA-RISE program through "Canleish" 101007653 (L.Ö.), and Olle Engkvist Foundation 196-0077 (Z.Z.). Dr. Tesfalem Welearegay is acknowledged for his help on the selectivity tests.

## Author contributions

Z.Z. conceived the idea and initiated the project. Q.H. and Z.Z. designed the experiments. Q.H. performed device fabrication and characterization, gas sensing measurements, simulation, data analysis, and modeling under the supervision of Z.Z. L.Ö. guided the gas sensor tests and analysis. Q.H. wrote the manuscript. P.S., L.Ö., and Z.Z. analyzed the data and revised the manuscript.

## Funding

## Competing interests

The authors declare no competing interests.
