## [Peer Review File · Nature Communications]

Nanotransistor-Based Gas Sensing with Record-High Sensitivity Enabled by Electron Trapping Effect in NanoparticlesREVIEWER COMMENTS

Reviewer #1 (Remarks to the Author):

The authors have presented a Palladium NP decorated double-gated SiNW FET with airgap that can exploit direct capacitive coupling and quantum mechanical tunneling between the conducting channel and Pd-NP for transduction. The authors also reported a high sensitivity with their proposed design. Overall the work has sufficient originality and scientific contribution to merit publication in Nature Communication. Therefore, the reviewer is suggesting acceptance of the article after mandatorily addressing the following concerns:

1. Please add the TCAD simulated-

(i) electrostatic potential variations by taking a cutline from side-gate to side-gate (through the channel) in SiNW-NAG, and from the back gate to the top of the channel (through the back gate insulator) in back-gate SiNW FET to the supplementary figure.3.

(ii) hole density variations by taking a cutline from side-surface to side-surface of the channel in SiNW-NAG, and from the back-surface to the top-surface of the channel n back-gate SiNW FET to the supplementary figure.6.

These will further enhance the clarity of comparative device electrostatics between the two designs.

1. The authors need to provide details about the TCAD simulation methodology including the specific TCAD tool used (Silvaco or Sentaurus), the physical models used, and how the transduction mechanism (H₂ off/H₂ on) is modeled in the TCAD environment by referring to the appropriate simulation-based literature.

2. The statement "The simulation results also show a higher response of the SiNW-NAG FET device (see Supplementary Section 6)" is lacking clarity. Please elaborate on the explanations, and include the changes suggested in the next comment to support the statement.

3. For a general reader please provide a detailed account of the essential physics of Pd workfunction modulation with the exposure in H₂, preferably through schematics (the authors may refer to this literature- <https://doi.org/10.1063/1.4775358>, <https://doi.org/10.1109/LENS.2020.2988589>).

4. For general readers, please include a brief discussion of why the Pd is preferred among other catalytic metals for work function modulated transduction for H₂ sensing.

5. It is interesting to see how the undesirable trapped charge density in the gate insulator region may affect the transduction mechanism. The authors are suggested to include at least a qualitative discussion or if possible include a TCAD simulation study in this context.

Reviewer #2 (Remarks to the Author):

1. **Complex Terminology:** The paper contains technical terms and concepts related to nanoscale FETs, electron tunneling, and signal transduction mechanisms. This complexity may pose a challenge for readers with a limited background in nanotechnology or semiconductor physics.

2. **Limited Comparison:** While the paper mentions the weaknesses of conventional FET-based sensors, a more detailed comparison with existing alternatives could enhance the paper's context and highlight the uniqueness of the proposed design.

3. **Selectivity Testing:** While the paper briefly mentions selectivity testing against CO and NO₂, a more in-depth exploration of potential cross-interference and the sensor's response to various gases would strengthen the evaluation of its practical utility.

4. **Application and Implementation Discussion:** The paper could benefit from a more detailed discussion on the potential real-world applications of the proposed sensor and the practical considerations for implementing such a design in different scenarios.

5. **Mismatch:** In page no 3 , mentioned about Buried SiO₂ while in supplementary paper it is mentioned as Buired

RESPONSE TO REVIEWERS' COMMENTS

Reviewer #1:

The authors have presented a Palladium NP decorated double-gated SiNW FET with airgap that can exploit direct capacitive coupling and quantum mechanical tunneling between the conducting channel and Pd-NP for transduction. The authors also reported a high sensitivity with their proposed design. Overall, the work has sufficient originality and scientific contribution to merit publication in Nature Communication. Therefore, the reviewer is suggesting acceptance of the article after mandatorily addressing the following concerns:

1. Please add the TCAD simulated-

(i) electrostatic potential variations by taking a cutline from side-gate to side-gate (through the channel) in SiNW-NAG, and from the back gate to the top of the channel (through the back gate insulator) in back-gate SiNW FET to the supplementary figure 3.

(ii) hole density variations by taking a cutline from side-surface to side-surface of the channel in SiNW-NAG, and from the back-surface to the top-surface of the channel in back-gate SiNW FET to the supplementary figure 6.

These will further enhance the clarity of comparative device electrostatics between the two designs.

Reply: The electrostatic potential and hole density variations along the cutlines are added into Supplementary Fig. 3 and 6, respectively, which further support the analysis of the direct capacitive coupling and enhanced signal of the SiNW-NAG FET device.

2. The authors need to provide details about the TCAD simulation methodology including the specific TCAD tool used (Silvaco or Sentaurus), the physical models used, and how the transduction mechanism (H_2 off/ H_2 on) is modeled in the TCAD environment by referring to the appropriate simulation-based literature.

Reply: The commercially available Sentaurus TCAD device simulator (version: S-2021.06-SP1) is used for the FET simulation. Mobility models include doping dependence, high-field saturation, and transverse field dependence. The intrinsic carrier concentration is determined with the silicon bandgap narrowing model OldSlotBoom. Shockley–Read–Hall model with doping-dependent lifetime is used for simulating recombination process.

In supplementary section 6, the H_2 transduction mechanism is simulated using TCAD tool with 4 nm thick SiO_2 passivation layer, in which the signal transduction is primarily ascribed to the capacitive coupling of the H_2 -induced PdNP work function change. The PdNP work function change is modeled in TCAD environment by setting a dipole layer at the interface between SiO_2 passivation layer and surrounding air region (the material of air region is selected as vacuum when building the device structure). The potential drop of the interface dipole layer is set at 50 mV in the simulation by choosing proper dipole density and interface permittivity.

The details of the TCAD simulation and corresponding references were added to Methods Section and Supplementary Section 6.

3. The statement "The simulation results also show a higher response of the SiNW-NAG FET device (see Supplementary Section 6)" is lacking clarity. Please elaborate on the explanations, and include the changes suggested in the next comment to support the statement.

Reply: The 1D profiles of hole density extracted from the TCAD simulation results are added to Supplementary Section 6, which clearly reveal the larger variation of hole density induced by H₂ in the SiNW-NAG FET device compared to the back-gate FET device.

4. For a general reader please provide a detailed account of the essential physics of Pd work function modulation with the exposure in H₂, preferably through schematics (the authors may refer to this literature- <https://doi.org/10.1063/1.4775358>, <https://doi.org/10.1109/LSENS.2020.2988589>).

Reply: The theory of the H₂-induced Pd work function modulation is illustrated at the beginning of the 'side and back-gate capacitive coupling' section (marked with green). A corresponding schematic is also inserted in Fig. 1c in the revised version.

5. For general readers, please include a brief discussion of why the Pd is preferred among other catalytic metals for work function modulated transduction for H₂ sensing.

*Reply: Pd is a suitable catalyst for H₂ sensing for several reasons. H is soluble in Pd and at 1 atm pressure and room temperature the H/Pd reach 0.7. Hydrogen dissociates and populates interstitial lattice sites when it adsorbs on Pd and diffuses into the bulk to form a Pd hydride phase. The formation and dissociation of palladium hydride are **reversible** processes. When hydrogen is absorbed into the Pd lattice it modifies the electronic properties (the H1s-Pd4d bonding), leading to a detectable shift of the work function as H (donor) hybridizes with Pd (acceptor). For modeling purposes, the sensing mechanism of a Pd sensor is typically described by the formation of a dipole layer as the interface becomes polarized (Lundstrom 1975). This dipole layer decreases the work function of metal. The work function of Pd is sensitive to changes in hydrogen across a **wide range of concentrations** and allows for precise measurement of hydrogen concentration (here, we demonstrate ultra-low ppb detection). Moreover, the H-induced change in work function of Pd is a **fast and selective** process, which is shown in Fig. 4c. Our device can readily discriminate H₂ from electrophilic (NO₂) and neutral molecules (CO). The small up-shift for NO₂ reported in Fig. 4c is likely due to NO₂ adsorption on the channel walls, while the insensitivity towards CO is the absence of charge transfer.*

A brief discussion has been added to the text to explain why Pd is the preferred catalytic metal for work function modulated H₂ sensing.

Reference:

*Lundstrom, I., Shivaraman, S., Svensson, C. & Lundkvist, L. A hydrogen-sensitive MOS field-effect transistor. Appl. Phys. Lett. **26**, 55–57 (1975).*

6. It is interesting to see how the undesirable trapped charge density in the gate insulator region may affect the transduction mechanism. The authors are suggested to include at least a qualitative discussion or if possible include a TCAD simulation study in this context.

Reply: Indeed, besides the PdNP traps located on the surface of SiO₂ passivation layer, there exists the intrinsic electron traps in the SiO₂ passivation layer (gate insulator region), which are mainly defects and Si dangling bonds at the Si/SiO₂ interface. The atomic hydrogen, forming after the H₂ dissociation at PdNP surface, can passivate the classical electron traps (Evans 1986) and thereby result in the electron transfer process like the PdNP traps. The existence of either the intrinsic electron traps, or the PdNP traps, can degrade the sub-threshold slope of FET, which in terms reflect the trap density. As shown in Fig. 2b, the slope for the PdNP-free FET (intrinsic electron traps only)

is much sharper than the one with the PdNP-coated FET (both intrinsic electron traps and PdNP traps), which indicates the density of the PdNP traps is significantly higher than that of the classical electron traps. Consequently, the electron transfer flow generated from the H₂ passivation of the intrinsic electron traps is negligible compared to the flow from the PdNPs. The transduction mechanism is dominated by the electron trapping effect of the PdNPs in our device.

Reference:

Evans, N. J., Petty, M. C. & Robert, G. G. Interface state effects in Pd-gate MOS hydrogen sensors. *Sensors and Actuators* **9**, 165–175 (1986).

Reviewer #2:

1. Complex Terminology: The paper contains technical terms and concepts related to nanoscale FETs, electron tunneling, and signal transduction mechanisms. This complexity may pose a challenge for readers with a limited background in nanotechnology or semiconductor physics.

Reply: The concepts, terms, and symbols used in the manuscript are critical to physically illustrate the advantages of the SiNW-NAG FET structure and the electron trapping effect. To assist the general readers to understand the highlights of our work, we plot the detailed schematics in Fig. 1c, Fig. 2c, and Fig. 2f using the accessible concepts (carrier distribution and energy band diagram) and symbols (potential: V , capacitance: C , and work function: Φ). Besides, we focus on displaying and analyzing the sensing performance of our sensors in the main text, leaving the detailed physics and math analysis in the supplementary information, which we believe will make reading of the main manuscript easier for a wider audience.

2. Limited Comparison: While the paper mentions the weaknesses of conventional FET-based sensors, a more detailed comparison with existing alternatives could enhance the paper's context and highlight the uniqueness of the proposed design.

Reply: The weakness of the reported ultrathin-channel and side-gate FET H₂ sensors is illustrated after the first paragraph of the 'side and back-gate capacitive coupling' section (marked with green). The unique merit of our SiNW-NAG FET over these devices is the direct coupling between the sensing material and the FET channel.

3. Selectivity Testing: While the paper briefly mentions selectivity testing against CO and NO₂, a more in-depth exploration of potential cross-interference and the sensor's response to various gases would strengthen the evaluation of its practical utility.

Reply: The work function of Pd is sensitive to changes in hydrogen across a wide range of concentrations. When hydrogen is absorbed into the Pd lattice it modifies the electronic properties, leading to a detectable shift of the work function as H (donor) hybridizes with Pd (acceptor). For modeling purposes, the sensing mechanism of a Pd sensor is typically described by the formation of a dipole layer as the interface becomes polarized. This dipole layer decreases the work function of metal by small addition of H. This sensitivity allows for precise measurement of hydrogen concentration. The H-induced change in work function of Pd is a fast and selective process, which is shown in Fig. 4c.

To address the reviewer's comment on the sensor's responses to various gases and practical utility, we have added data for ammonia (NH₃) in the revised manuscript. The motivation for adding NH₃ as a test gas, is three-fold: 1) It

extends our tested gases also to include a classical nucleophilic molecule, 2) NH₃ exposure under our conditions is known to lead to NH₃ dissociation and PdN_x formation with small amounts of N₂ and H₂O products, and very little NO_x species (Dunn 2019), thus creating a dipole layer (as for H₂, but smaller), and 3) it is a technically important gas. Ammonia has e.g. been suggested as a potential storage medium for hydrogen that is easier to transport, less costly to produce and has a higher energy density than that of compressed hydrogen gas. Our device can readily discriminate between H₂ and electrophilic (NO₂) and neutral (CO) molecules, while the sensitivity for NH₃ is almost 100 times lower. The small up-shift for NO₂ reported in Fig. 4c is likely due to NO₂ adsorption on the channel walls, while the insensitivity towards CO is the absence of charge transfer. It should also be noted that the sensor measurements were conducted in synthetic air, which brings the “testing” closer to real-world applications.

A brief discussion has been added in the text to explain why Pd is the preferred catalytic metals for work function modulated H₂ sensing. A sentence on sensor measurement conditions (synthetic air carrier gas) have been added in the revised main text to highlight the potential real-world application of the device. The test conditions are also articulated more clearly in the Methods Section.

Reference:

*Dann, E.K. et al. Structural selectivity of supported Pd nanoparticles for catalytic NH₃ oxidation resolved using combined operando spectroscopy. Nat. Catal. **2**, 157–163 (2019).*

4. Application and Implementation Discussion: The paper could benefit from a more detailed discussion on the potential real-world applications of the proposed sensor and the practical considerations for implementing such a design in different scenarios.

Reply: Considering the performance of our sensor, environmental and leakage monitoring in vehicles, buildings and process industry would be representative potential real world applications. An important advantage of our sensors is their small power consumption that promises large scale and wireless deployment. The characteristics demanding further optimization include the long-term stability and wide range operation temperature, as well as gas mixture responses and humidity. A brief discussion is added at the end of the ‘kinetics and selectivity’ section (marked with green).

5. Mismatch: In page no. 3, mentioned about Buried SiO₂ while in supplementary paper it is mentioned as Buired.

Reply: The typo in Supplementary Fig. 6 is corrected in the revised version.

REVIEWERS' COMMENTS

Reviewer #1 (Remarks to the Author):

The authors have satisfactorily addressed most of the concerns raised by the reviewer, the paper may be accepted for publication with the following mandatory minor revisions:

1. Give reference for Sentaurus TCAD and its physics based models used in this work.
2. For a more comprehensive understanding of the general reader, the recent TCAD based theoretical reports on Pd-based catalytic metal gated FET (TFET) sensors needs to be discussed. In this context, the authors should include the following articles in their literature review-

<https://10.1109/LSENS.2020.2988589>
<https://doi.org/10.1109/JSEN.2021.3072476>
<https://doi.org/10.1007/s12633-022-01829-x>
<https://doi.org/10.1007/s12633-022-02103-w>
<https://10.1109/LSENS.2023.3272394>
<https://doi.org/10.1007/s12633-022-02242-0>
<https://doi.org/10.1002/adts.202301031>

3. For the comment "Limited Comparison: While the paper mentions the weaknesses of conventional FET-based sensors, a more detailed comparison with existing alternatives could enhance the paper's context and highlight the uniqueness of the proposed design."

The response is not adequate. The authors are strongly suggested to further elaborate the comparative performance analysis with previous reports.

RESPONSE TO REVIEWERS' COMMENTS

Reviewer #1 (who also evaluated the concerns of Reviewer #2):

1. Give reference for Sentaurus TCAD and its physics based models used in this work.

Reply: The reference for the TCAD simulation is added to the main text as reference 52.

2. For a more comprehensive understanding of the general reader, the recent TCAD based theoretical reports on Pd-based catalytic metal gated FET (TFET) sensors needs to be discussed. In this context, the authors should include the following articles in their literature review-

<https://doi.org/10.1109/LSENS.2020.2988589>

<https://doi.org/10.1109/JSEN.2021.3072476>

<https://doi.org/10.1007/s12633-022-01829-x>

<https://doi.org/10.1007/s12633-022-02103-w>

<https://doi.org/10.1109/LSENS.2023.3272394>

<https://doi.org/10.1007/s12633-022-02242-0>

<https://doi.org/10.1002/adts.202301031>

Reply: The discussion of TCAD simulation is added to the main text in the third paragraph on page 3 (marked with yellow) and the relevant papers are referred to.

3. For the comment "Limited Comparison: While the paper mentions the weaknesses of conventional FET-based sensors, a more detailed comparison with existing alternatives could enhance the paper's context and highlight the uniqueness of the proposed design."

The response is not adequate. The authors are strongly suggested to further elaborate the comparative performance analysis with previous reports.

Reply: More discussion of existing devices is added to the main text in the fourth paragraph on page 3 (marked with yellow).